# Feature representation in analysing childhood vaccination defaulter risk predictors: A scoping review of studies in low-resource settings

**Eliezer Ofori Odei-Lartey**[1,2☯*], **Stephaney Gyaase**[2☯], **Solomon Nyame**[2☯], **Dominic Asamoah**[1☯], **Kwaku Poku Asante**[2☯], **James Ben Hayfron-Acquah**[1☯]

**1** Department of Computer Science, Kwame Nkrumah University of Science and Technology, Kumasi, Ashanti Region, Ghana, **2** Kintampo Health Research Centre, Research and Development Division of Ghana Health Service, Kintampo, Bono East Region, Ghana

☯ These authors contributed equally to this work.
\* eliezer.lartey@kintampo-hrc.org

## Abstract

Childhood vaccination saves millions of lives yearly, yet over a million children in low-and middle-income countries die from vaccine-preventable diseases each year. Predicting childhood vaccination defaulter risk with analytical models requires understanding how to represent different individual demographics, community structures, and environmental factors that feed input data. This review explores features for analysing childhood vaccination defaulter risk in low-resource settings with a focus on feature encoding, engineering and representation. Articles published from 2018 to January 2025 were searched using PubMed, Google Scholar, ACM Digital Library, and references from the searched articles. Search was limited to low- and middle-income countries, focusing on African countries. We included studies that utilised either statistics or machine learning for analysis. Of the 4,174 articles retrieved, 55 were eligible, 41 were then excluded after full-text review, and 4 were added from references. Cross-cutting features included maternal education and health service utilisation. Novel features included community rates of poverty, maternal education and maternal unemployment. Variations in encoding strategies, engineering techniques and feature representation were marginal. Categorical data were mainly encoded as binary inputs, while features with high dimensionality like socio-economic status were condensed by using principal component analysis. A review of existing feature representations can serve as a feature construction reference to improve the exploitation of machine learning techniques within the context of childhood vaccination defaulter risk prediction. Future studies can exploit other representations different from binary encoding, like frequency encoding, to introduce elements of weighting into multi-categorical features.

**Data availability statement:** The data supporting the findings of this scoping review are primarily sourced from publicly available literature identified through comprehensive database searches and are therefore accessible through the cited publications. To enhance transparency and reproducibility, we developed a structured data extraction file, capturing key study-level variables: country context, study design, predictors analysed, encoding strategies used, analytical methods applied, and outcome definitions. This dataset is available as a supplementary Excel file.

**Funding:** The author(s) received no specific funding for this work.

**Competing interests:** The authors have declared that no competing interests exist.

## Author summary

Childhood vaccination saves millions of lives every year, but more than a million children in low- and middle-income countries die each year from vaccine-preventable diseases. To predict childhood vaccine defaulter risk using analytical models, it is important to understand how to effectively represent the features and variables that constitute the input data. This review examined features for analysing childhood vaccine defaulter risk in low-resource settings, with an emphasis on feature encoding, engineering, and representation. Articles published between 2018 and January 2025 were searched using PubMed, Google Scholar, the ACM Digital Library, and references from the searched articles. The search concentration was on articles emanating from African countries. We considered articles that used statistics or machine learning for analysis. Of the 4,174 articles retrieved, 55 were eligible, 41 were removed following full text review, and 4 were added from references. Features related to maternal education and the use of health care services were cross-cutting. Novel features included community-based poverty rates, community maternal literacy rates, and community maternal unemployment rates. Differences in feature encoding, engineering, and representation were marginal among studies. Categorical data were mostly encoded as binary data, whilst high-dimensional features determining socioeconomic status were reduced to a single-point index using principal component analysis. This review provides a possible feature representation reference for studies that intend to exploit machine learning techniques in the analysis and prediction of childhood vaccination defaulter risk. Future studies can incorporate representations other than binary encoding, such as frequency encoding, to incorporate weighting aspects into multi-categorical characteristics.

## Introduction

The World Health Organisation (WHO) estimates that childhood vaccination prevents approximately 2–3 million child deaths per year [1]. Yet, the number of children who die from vaccine-preventable deaths in low-resource environments remains significant. A study by [2] estimates that low-and-middle-income countries (LMICs) account for about one million childhood deaths from vaccine-preventable diseases each year. Chandir and colleagues [3] cited studies showing high coverage rates on earlier vaccinations and lower rates on vaccines administered later in the vaccination schedule. This was still confirmed in a more recent study by Nantongo and colleagues [4], indicating that the defaulting rate increased by infant age. To realise the full impact of vaccination in LMICs, it is imperative to consider effective strategies for improving outreach and vaccination uptake among children at risk of defaulting from this critical intervention programme. Without an effective strategy, many LMICs may not meet the target of at least 90% vaccination coverage, which the World Health Organisation

Immunisation Agenda 2030 has earmarked as the per-country requirement to help eradicate or reduce avoidable deaths caused by diseases for which vaccines are available [5].

One increasing research interest is to exploit advanced analytics that can be used to effectively and efficiently identify populations or clusters of individuals with high defaulter risk, allowing for more targeted interventions, with machine learning being a significant area of focus. Machine learning (ML) is a novel technique that emulates human intelligence to predict an outcome by learning from historical data [6]. In public health, the contribution of ML techniques to unravel insights and predict from complex data is well documented [7–9]. There were heightened research interests, following the COVID-19 pandemic, in using machine learning models to predict disease spread patterns [10]. Machine learning relies heavily on feature relevance in informativeness, which largely depends on how well the features have been represented. The features, which are sometimes called predictors, serve as the input data. They must be appropriately represented to ensure the functionality and accuracy of machine learning models.

Features for the analysis of childhood vaccination defaulter risk in low-resource settings relate to various demographic characteristics, health behaviour patterns, geographical disparities, socio-economic conditions, and environmental factors. Extensive reviews have already been done elsewhere [11–14] to highlight relevant features and their comparative significance in predicting childhood vaccination defaulter risk in low-resource settings. However, the granularities of how they were encoded, represented or engineered for the models are often marginally mentioned. Yet, these factors must be accurately encoded and represented to serve as valuable inputs for analytical models. An elaborate review of how various features have been represented can serve as a reference for emerging researchers on the best representations for suitable machine learning models in the wider context of public health. We conducted this review to unravel how key childhood vaccination defaulter risk features in various studies were represented for analysis. In conducting this review, we neither confirmed nor refuted the appropriateness of features nor established relative quality among the features identified. The review was to capture and synthesise up-to-date evidence essential for understanding the encoding, representation and engineering of key features that may influence the accuracy of machine learning models.

## Methods

### Basic definitions

We refer to a defaulter in all three possible scenarios where a child either missed out on any vaccination for age, was delayed in receiving any vaccine per standard schedule or was never vaccinated (zero-dose) [15]. Also, we refer to low-resource settings as low-and-middle-income countries as classified by the Organisation for Economic Co-operation and Development (OECD) as of 19th February 2025. We also refer to features or predictors as the independent variables used in various analysis models to predict an outcome. The terms settings and environments are used interchangeably. In this study, we also interchange the terms defaulter risk, adherence and compliance risk. The terms predictors and features are also used interchangeably.

### Search strategy

An up-to-date literature search up to 18 January 2025 was conducted electronically using different search engines, including PubMed, Google Scholar and ACM Digital Library, and references from searched articles. Search terms and keywords used are described below.

PubMed Central: ((predictors OR parameters) AND (childhood vaccination OR childhood immunisation) AND (low resource OR developing) AND (settings OR environments)) OR ((predictors OR parameters) AND (childhood vaccination OR childhood immunisation) AND (low resource OR developing) AND (settings OR environments) AND (machine learning OR artificial intelligence))

Google Scholar: ((predictors OR parameters) AND (childhood vaccination OR childhood immunisation) AND (low resource OR developing) AND (settings OR environments)) OR ((predictors OR parameters) AND (childhood vaccination OR childhood immunisation) AND (low resource OR developing) AND (settings OR environments) AND (machine learning OR artificial intelligence))

ACM Digital Library: ((predictors OR parameters) AND (childhood vaccination OR childhood immunisation) AND (low resource OR developing) AND (settings OR environments)) OR ((predictors OR parameters) AND (childhood vaccination OR childhood immunisation) AND (low resource OR developing) AND (settings OR environments) AND (machine learning OR artificial intelligence)).

The database engines were searched, and all searches were limited to up-to-date articles between 2018 and January 2025 to ensure the inclusion of the most recent and relevant studies reflecting current trends in childhood vaccination defaulter risk analysis. The search was restricted to studies conducted in the English language to maintain consistency in data interpretation and avoid potential translation biases. Additionally, the focus on low-resource environments, and particularly on African countries, provides insights specific to settings where vaccination uptake challenges are well-known.

### Eligibility criteria

Articles reporting on the predictors of childhood vaccination defaulter risk in low-resource environments were included in this review. In terms of study design, only quantitatively driven studies were included. As such, studies with qualitative designs were excluded. Interest was not limited to articles that used machine learning techniques for analysis. Also, articles that used classical statistical methods (e.g., logistic regression, multivariate analysis) were included, as they equally require a high level of feature encoding, engineering and representation. Regarding context, particular interest was given to articles involving data from African settings, while those that utilised data from countries classified as high-income by the OECD were excluded. Additionally, excluded articles included those focusing on special groups (e.g., HIV patients or specific religious groups) as well as studies that concentrated on predictors associated with the COVID-19 outbreak.

### Screening and selection

The search results were compiled in the Rayyan web app [16] to facilitate the screening process. During the initial screening, duplicates were removed. After removing duplicates, an initial assessment of article relevance was conducted by two independent reviewers, EO and SG. Using the Rayyan web app, the reviewers annotated relevant articles as "Included", rejected articles as "Excluded", and articles requiring further review as "Maybe". Any disagreements regarding article selection were resolved through discussions between the two reviewers and a third reviewer, SN. A final full-text review process was performed on the relevant articles to further assess their relevance in terms of context, outcomes, and design.

### Data charting and analysis

The Microsoft Excel spreadsheet was used to organise findings from this review. Key themes employed during synthesis included study characteristics, data sources, significant predictors, methods of analysis, and outcomes. To enhance transparency and reproducibility, we developed a structured data extraction file, capturing key study-level variables: country context, study design, predictors analysed, encoding strategies used, analytical methods applied, and outcome definitions. This dataset is available as a supplementary Excel file (S1 Data).

## Results

### Study selection

Out of 4174 articles obtained, 55 were eligible for the scoping review. Subsequently, 41 of the eligible studies were excluded for various reasons related to context, outcome and study design. In addition to the 14 eligible studies, four (4)

new studies from the reference of eligible studies were identified and included. Fig 1 is the PRISMA scoping review flow chart designed using tools from Haddaway and colleagues [17].

## Study characteristics

The studies used a wide range of statistical methods and machine learning techniques to evaluate features of childhood vaccination adherence behaviours and uptake in low-resource settings. A total of 13 out of the 18 studies used secondary data, while 5 studies used primary data from cross-sectional designs. Most studies analysed data from African countries: Uganda, Ethiopia, South Africa, Ghana, Nigeria, and the Democratic Republic of Congo. Two other studies analysed data from Bangladesh, while another analysed data from 92 LMICs. In all studies, the prediction of vaccination adherence was to either classify full versus zero-dose, complete versus incomplete dose, or timely versus delayed dose. Accordingly, a binary outcome was predicted using different methods including binary logistic regression, decision tree analysis [4,18], Naïve Bayes [4,18,19], random forest [4,18,19], support vector machines [4], neural networks [19,20], and deep learning [21]. In Table 1, we summarise the predictors identified in the articles. We also describe the methods used to predict the outcome of interest. Supplementary narratives of Table 1 are provided in S1 Appendix.

## Analytical summary of reviewed studies

Across the 18 reviewed studies, over 60 unique predictor variables were identified. Of these, the most frequently occurring across studies was maternal education (n = 14 studies). ANC visits, place of delivery and place of residence had equal occurrence (n = 9 studies). The indicator of socio-economic status (wealth index) also occurred in 8 studies, similar to maternal age (n = 8 studies). Other frequently used predictors were child-related demographics such as the sex of the child (n = 6 studies), birth order of child (n = 5 studies) and the age of the child (n = 4 studies). Table 2 presents the top 15 most common predictors across reviewed studies.

The reviewed studies employed a range of statistical and machine learning techniques to analyse predictors of childhood vaccination defaulter risk. Table 3 provides a summary of the modelling methods used across the reviewed studies and the studies that leveraged the models.

Binary logistic regression was the dominant statistical modelling method (n = 13) used among the studies reviewed, though other statistical methods such as multivariate analysis and basic descriptive summaries were used. Studies employing machine learning methods (n = 7) most commonly used ensemble techniques such as Random Forests (n = 3), different types of Gradient Boosting Machines (n = 3), or hybrid multilayer perceptrons (n = 3). These were typically applied to large secondary datasets, often with more than 5,000 observations. We also analysed the main feature engineering and representation strategies used in the studies reviewed. Results on the encoding strategies and frequency of occurrence across the studies are presented in Table 4.

From the results, one-hot binary encoding was the most cross-cutting strategy used for feature representation (n = 16 studies). A total of 10 studies further applied feature engineering beyond binary categorisation. Wealth index was the predominant predictor to which dimensionality reduction was applied (n = 8 studies), whiles two other studies applied binary encoding techniques to engineer novel indicators. Acharya and colleagues [32] engineered community level indicators of maternal education and media exposure rates into a binary vector representing low and high rates, whiles Santos and colleagues [26] created a binary representation from three distinct predictors for decision tree analysis.

## Classification of features

Features can be classified into two broad groups of individual/household and community/environmental factors. These categories can be further divided into six subcategories: socio-demographics, economic status, knowledge and behaviour, community and social structures, physical access and mobility, and environmental and climate conditions. Additionally, we noted the possibility of composite predictors that resulted from the combined effects of two or more predictors (Fig 2).

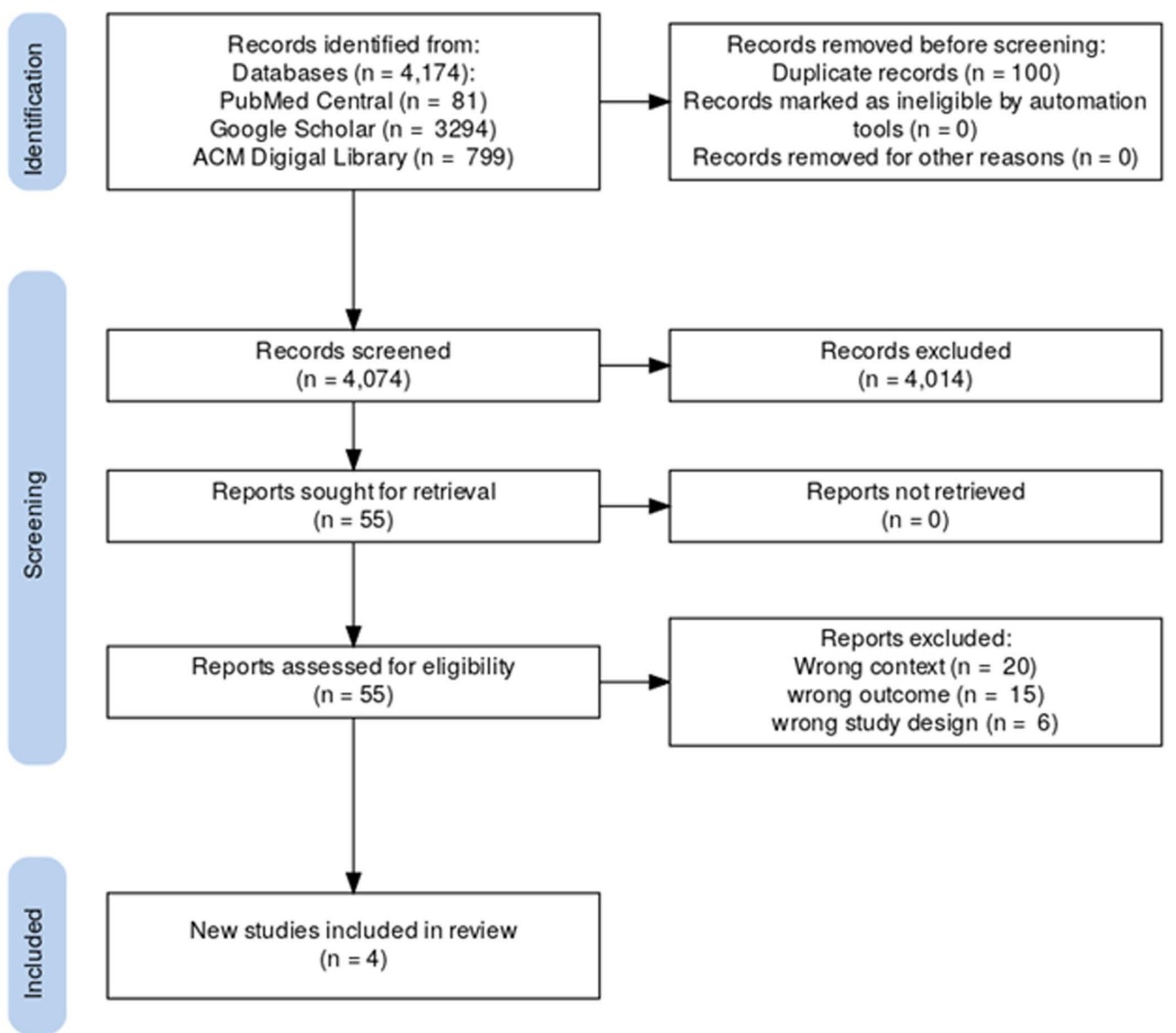

**Fig 1. The PRISMA flow chart diagram for this scoping review.** A flow diagram depicting the number of records identified, screened, excluded, and included in the final scoping review, following the PRISMA 2020 reporting guideline.

## Individual/household predictors

Basic individual demographic features like age, sex, education, marital status, religious affiliation and employment status were present in almost all studies. Additional features, such as reminder frequency and media exposure, were also

**Table 1. A table showing publications selected in association with routine childhood vaccination uptake.**

| Authors | Year | Data used | Methods of Analysis | Significant Predictors | Outcome |
|---|---|---|---|---|---|
| Nantongo, B., et al. [4] | 2024 | Secondary data from DHS of 8932 children aged 0 – 59 months, Uganda | Principal Component Analysis, k-Nearest Neighbours, Decision Trees, Random Forests (RFs), Support Vector Machine (SVM), Naïve Bayes, Logistic Regression (LR), XGBoost, Adoptive Boosting, and Gradient Boosting | Birth year of mother, language, place of residence, water source, ethnicity, maternal education, number of under-5 children, age of household head, maternal mobility, wealth index, age at first birth, maternal age, maternal employment, age of child, ANC visits, place of delivery, status of previous vaccination. | Receipt of complete vaccination for age |
| Hasan, M., et al. [18] | 2021 | Secondary data from DHS of about 20250 children aged 12 – 23 months, Bangladesh | Gaussian Naive Bayes (GNB), Bernoulli Naive Bayes (BNB), Decision Tree (DT), Random Forest (RF), XGBoost (XGB), LightGBM (LGB), ensemble of ML models | Settlement type, place of residence, maternal education, religious affiliation, media exposure, wealth index, paternal education, maternal employment, sex of child, sex of household head, maternal age, household size, age at birth, birth order, ANC visits | Receipt of first dose of measles vaccination |
| Demash, A., et al. [19] | 2023 | Secondary data from DHS of 1617 children aged 12 – 23 months, Ethiopia | Naïve Bayes, PART, logistic regression, multilayer perceptron, J48, logit boost, random forest, and AdaBoost. | wealth index, maternal education, maternal age, place of residence, sex of child, age of child, birth interval, birth order, sex of household head, ANC visit, place of delivery, maternal employment, media exposure | Receipt of complete vaccination for age |
| Biswas, A., et al. [20] | 2023 | Secondary data from DHS of ~95,000 children aged 12 – 23 months, India, Mali and Nigeria | Cost-sensitive Ridge Classification, Nearest Neighbour, Multilayer Perceptron | Place of residence, place of delivery, ANC visits, age of the child, child's birth order, sex of child, first pregnancy, wealth index, maternal education | Zero-dose status |
| Mohanraj G., et al. [21] | 2020 | Secondary data from DHS of 5057 children aged 12 – 23 months, India | Rank-Based Multi-Layer Perceptron hybrid deep learning framework, Deep Soft Cosine Semantic and Ranking SVM based model, Decision Tree, Naïve Bayes, Linear Regression | Maternal education, maternal employment, vaccination card availability, availability of health insurance scheme, paternal employment, sex of child, ANC visits | Receipt of complete vaccination for age |
| Abatemam, H., et al. [22] | 2023 | Primary data of 422 children aged 0 – 23 months, Ethiopia. | Binary Logistic Regression | Maternal education, settlement type, knowledge of benefits, reminders | Receipt of complete vaccination for age |
| Abegaz, M., et al. [23] | 2023 | Primary data of 441 children aged 12 – 23 months, Ethiopia. | Binary Logistic Regression | Maternal age, place of delivery, knowledge of benefits, travel time, marital status, maternal education, ANC visits, reminders | Receipt of complete vaccination for age |
| Muhoza, P., et al. [24] | 2023 | Secondary data from IGD of 1522 children aged 18 – 35 months, Ghana | Binary Logistic Regression | Maternal age, reminders, settlement type, birth order, place of residence | Receipt of second year life dose of measles 2 and Meningococcal Serogroup A vaccinations in Ghana |
| Aheto, J., et al. [25] | 2022 | Secondary data from DHS of ~21000 children aged 12 – 35 months, Nigeria | Binary Logistic Regression | Vaccination card availability, maternal age, received vitamin A, maternal employment, maternal education, religious affiliation, internet/phone access, ethnicity, bank account, livestock, travel time | Receipt of three different vaccines as three different outcomes |
| Santos, T., et al. [26] | 2021 | Secondary data from DHS and IGD of 210, 509 children aged 12 – 23 months, 29 LMICs | Classification and Regression Tree | ANC visits, place of delivery, maternal tetanus vaccination status, wealth index, settlement type, maternal education, a triple predictor (no ANC visit & home delivery & no tetanus) | Zero-dose vaccination |
| Touré, A., et al. [27] | 2021 | Primary data of 380 children aged 0 – 59 months, Guinea | Binary Logistic Regression | Vaccination card availability, ANC visits, birth order, sex of child, place of delivery, place of residence, ill before schedule, knowledge of benefits | Receipt of complete vaccination for age |

*(Continued)*

**Table 1.** (Continued)

| Authors | Year | Data used | Methods of Analysis | Significant Predictors | Outcome |
|---|---|---|---|---|---|
| Budu, E., et al. [28] | 2020 | Secondary data from DHS of 5119 children aged 12 – 23 months, Ghana | Multivariate Statistical Analysis Binary Logistic Regression | Maternal education, religious affiliation, settlement type, ethnicity, parity, wealth index, place of residence | Receipt of complete vaccination for age |
| Jama, A. [29] | 2020 | Primary data of 315 children aged 11 – 24 months, Somalia. | Descriptive Statistics (frequency, mean and standard deviation) | Maternal education, place of delivery, travel time | Receipt of complete vaccination for age |
| Acharya, K., et al. [30] | 2019 | Secondary data from DHS of 4330 children aged 12 – 23 months, India | Logistic Regression | Place of residence, sex of child and maternal education | Receipt of complete vaccination for age |
| Adamu, A., et al. [31] | 2019 | Primary data of 675 children aged 0 – 23 months, Nigeria | Logistic Regression Markov Chain Monte Carlo | Travel time, number of vaccinators in facility, birth order, age of child, type of health facility, sex of caregiver, age of caregiver | Zero-dose vaccination |
| Acharya, P., et al. [32] | 2018 | Secondary data from DHS of 3366 children aged 12 – 32 months, DRC | Binary Logistic Regression | Sex of child, place of delivery, preceding birth space of +/- 24 months, age of father, maternal education, ANC visits, PNC visits, religious affiliation, maternal employment, media exposure, wealth index, maternal autonomy, employment status of father, settlement type, travel time, community media exposure rate, community poverty rate, community ANC visit rate, community facility delivery rate, community maternal unemployment rate, community maternal literacy rate, community PNC visit rate. | Receipt of complete vaccination for age |
| Asuman, D., et al. [33] | 2018 | Secondary data from DHS of 6533 children aged 12 – 59 months, Ghana | Binary Logistic Regression Econometric analysis techniques | Age of child, place of delivery, maternal age, maternal education, marital status, religious affiliation, maternal employment, maternal health insurance status, wealth index | Receipt of complete vaccination for age |
| Sheikh, N., et al. [34] | 2018 | Secondary data from DHS of 1631 children aged 12 – 23 months, Bangladesh | Binary Logistic Regression | Birth seasons, maternal employment, source of water, toilet facilities, place of residence, household size, maternal education, travel time, wealth index, maternal age, hygienic toilet facilities | Receipt of complete vaccination for age Delays in the receipt of vaccines |

**IGD**: Investigator-generated data, **ANC**: Antenatal care during pregnancy, **PNC**: perinatal care, **DHS**: District Health Survey.

identified. Regarding socio-economic status, ownership of specific household assets, characteristics of housing, and access to essential utilities were used to compute a wealth index feature [4,18,19,26,28,32–34]. The wealth index computation is a common method used to measure socio-economic status to mitigate the arduous task of acquiring accurate income data [35]. Predictors associated with health behaviour or health practices, such as ANC attendance, place of delivery, PNC attendance, possession of health insurance, and incidence of child illness before vaccination, were also present [4,18–21,23,26,27,32]. Also, Santos and colleagues [26] identified a mother's receipt of the tetanus vaccine during pregnancy as a significant predictor of childhood vaccination defaulter risk. Predictors linked to household factors included household size, the sex of the household head, and the relationship of the child to the caregiver were presented by Hasan et. al. [18] and Sheikh et al. [34].

## Community/environmental predictors

In terms of community structures, Acharya, P. et al. [32] listed maternal autonomy in decision-making as a key feature in their analysis. Authors added new features, namely, community poverty rate, community maternal literacy rate, community

**Table 2. Top 15 predictors identified across reviewed studies.**

| Predictor | Occurrence (Number of Studies) |
| --- | --- |
| Maternal Education | 14 |
| Antenatal care visits | 9 |
| Place of delivery | 9 |
| Place of residence | 9 |
| Wealth Index | 8 |
| Maternal Age | 8 |
| Maternal Employment | 7 |
| Settlement Type | 6 |
| Travel time to vaccination point | 6 |
| Sex of child | 6 |
| Birth order of child | 5 |
| Age of child | 4 |
| Reminders | 3 |
| Media Exposure | 3 |
| Vaccination Card Availability | 3 |

**Table 3. Modelling approaches across reviewed studies.**

| Modelling method | Number of Studies |
| --- | --- |
| Logistic Regression | 13 |
| Naïve Bayes | 4 |
| Decision Tree | 4 |
| Random Forest | 3 |
| Gradient Boosting Machines | 3 |
| Multilayer Perceptron | 3 |
| SVM | 2 |
| CART | 1 |
| Hybrid Deep Learning | 1 |
| Others (econometrics, multivariate) | 2 |

**Table 4. Trends in feature encoding strategies across studies.**

| Encoding Strategy | Number of Studies |
| --- | --- |
| Binary Encoding | 17 |
| Principal Component Analysis (PCA) | 8 |
| One-hot encoding of multiple indicators | 1 |
| Binirisation of community-level rates | 1 |
| Not Reported | 1 |

ANC attendance rate and community PNC attendance rates. Some predictors related to environmental and climatic conditions emerged during the review. With regards to environmental and climatic determinants, Sheikh and colleagues [34] identified that the hygienic conditions of essential utilities (water and toilet facilities) and season of birth were significant predictors. With regard to physical access, almost all studies identified place of residence, settlement type and travel

PLOS Digital Health

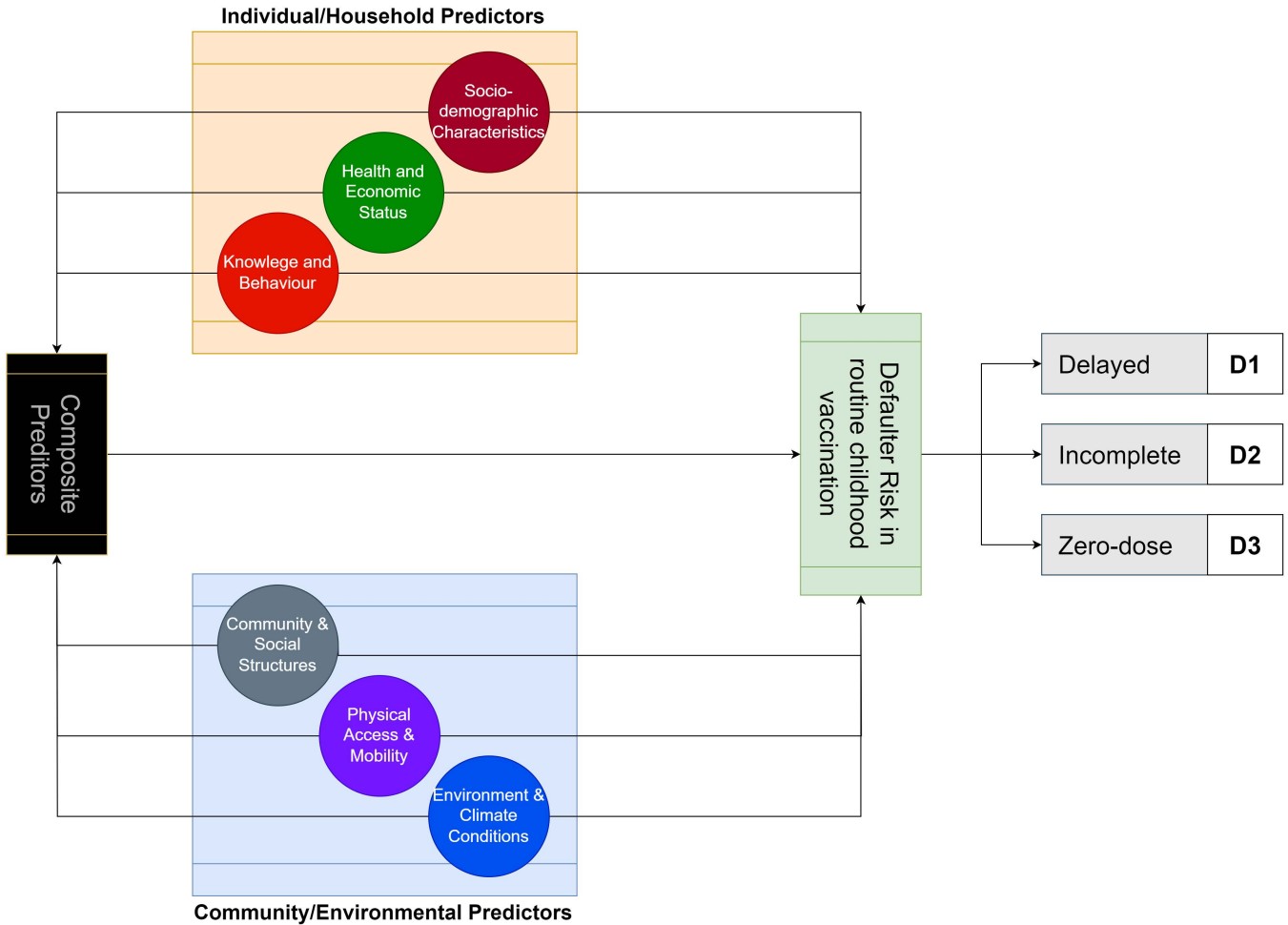

**Fig 2. Model for the review of childhood vaccination defaulter risk predictors.** The model diagram depicts the conceptual notion of grouping features into two broad groups of individual and community-level factors. The figure also depicts the complex interaction existing between these factors that influence childhood vaccination adherence.

time to the vaccination point as important predictors. In addition, Nantongo and colleagues [4] identified maternal mobility (computed as the frequency of travel within 12 months) as a significant predictor of childhood vaccination defaulter risk. Table 5 summarises the definitions of predictors and how they were characterized in studies.

### Feature encoding, engineering and representation

To better contextualise how predictors were prepared for analysis across the reviewed studies, Table 6 provides a consolidated mapping of key features to which various encoding strategies were applied, their respective encoding strategies, analytical models, and outcome definitions. The predictors most frequently used were maternal education, antenatal care (ANC) visits, place of delivery, wealth index, and settlement type. Features were represented as binary responses in over 90% of the studies. In several studies multi-response categorical features were converted into dichotomous responses. Sheikh and colleagues [34] represented place of birth as home or health facility, birth size as normal and small, type of drinking water as improved and non-improved, and cooking fuel types as clean and polluting. Acharya and colleagues [32] also used community-level predictors like community poverty rate, community maternal unemployment rate, and

PLOS Digital Health

**Table 5. Summary of definitions of predictors and how they were characterized in studies.**

| Category | Decription of predictors |
| --- | --- |
| Individual/Household Predictors | |
| Socio-demographics | Maternal age was defined in terms of the age of the mother at the date of delivery and was characterised as different age groupings. [4,18,19,23–25,33,34]. Options for the marital status of the mother included married, single, living together, divorced/separated, and widowed [23,33]. Maternal education was defined in terms of the highest formal education level the mother had attained. [4,18,19,21–23,25,26,28–30,32–34]. Place of residence refers to the location of mother and child, which was based on administrative and regional boundaries [4,18,20,24,27,30,34]. Water source referred to the main source from which the mother or household collected water for drinking and cooking [4,34]. Ethnicity referred to the ethnic affiliation of the mother in the country's cultural context [4,25,28]. Maternal employment referred to whether or not the mother was economically employed or engaged in a commercial activity [4,18,19,21,25,32–34]. Household size was defined as the number of people that made up the household of the mother and child [18,34]. Characteristics of the household head included the sex of the household head [18,19] and the age of the household head [4,18]. Characteristics of the child used as predictors included the age of the child at the time of data collection, sex of the child (male or female), and birth order [18,19,24,27,31]. Also, Hasan, M., et al. [18] and Mohanraj G., et al. [21] included predictors such as paternal age and paternal employment as predictors. |
| Economic status | Nantongo, B., et al. [4] computed a single-point wealth index from several individual or household indicators of wealth. The studies used the Principal Component Analysis (PCA) method to compute the index, which was represented as quintiles. Aheto, J., et al. [25] analysed three indicators of economic status as separate predictors. This included whether or not the mother had access to or owned a bank account, whether or not the mother had access to the internet or a mobile phone, and whether or not the household owned livestock. |
| Knowledge and behaviour | ANC visits feature was characterised as the number of times the mother attended antenatal care during pregnancy. Acharya, P. et al. [32] indicated the use of a binary representation of whether or not the mother completed at least four antenatal visits. PNC visits followed a similar representation of the number of times a mother and child attended postnatal care after delivery. Also, Nantongo, B., et al. [4]; Abegaz, M., et al. [23]; Biswas, A., et al. [20]; Santos, T., et al. [26]; Touré, A., et al. [27]; Jama, A. [29]; Acharya, P., et al. [32]; and Asuman, D., et al. [33] included a place of delivery feature, which referred to the place the mother delivered the child. For the many studies that included this predictor, a distinction between institutional and home delivery was mainly represented. For studies where the completion of a specific vaccine was the outcome of interest, a feature we refer to as previously vaccinated indicated whether the child had received or missed a preceding vaccine. Another predictor used by Abatemam, H., et al. [22]; Abegaz, M., et al. [23]; and Muhoza, P., et al. [24] indicated whether or not the mother/carer received reminders about vaccination dates. This included mothers receiving reminders via a device about their next vaccination date or being informed by vaccinators about the next vaccination date of the child. Abatemam, H., et al. [22]; Abegaz, M., et al. [23]; and Touré, A., et al. [27] included a maternal knowledge feature to indicate whether or not the mother had knowledge about the benefits of childhood vaccinations. Some studies included a predictor about birth interval to indicate whether or not the pace between a previous live birth and the child of interest was small, with a ± 24 months threshold. Other predictors classified under knowledge included whether or not the mother had media exposure through sources such as radio and/or television [18,19,32]. Predictors classified under behaviour included parity as at delivery, whether or not mother had a valid health insurance cover, and whether or not the mother had a document containing the vaccination information of the child, whether or not the mother had received the maternal tetanus vaccination during pregnancy. and whether or not the child took ill before the schedule for the vaccination dose. |
| Community/Environmental Predictors | |
| Community and Social Structures | Acharya, P., et al. [32] included seven (7) novel community-based predictors: community poverty rate, community maternal literacy rate, community ANC attendance rate, community PNC attendance rate, community facility delivery rate, community maternal unemployment rate, and community media exposure rate. These rates were represented by authors as high or low based on a defined threshold. Authors also included a feature for maternal autonomy which indicated whether or not the mother was involved in financial and/or health care decisions. |
| Physical access and mobility | In terms of physical access, a feature for settlement type was defined to indicate whether the mother and child lived in rural and urban settlements. Also, a feature related to maternal mobility was also included by Nantongo, B., et al. [4], represented as the frequency of travel of the mother and child in the last 12 months. A travel time predictor was also defined in [23,25,29,31,32,34] as the time taken (estimated by the mother) to reach the vaccination point. |
| Environment and climate conditions | Sheikh, N., et al. [34] introduced a birth season feature to represent the season in which the child was delivered in Bangladesh. Authors also included water and toilet hygiene features to respectively represent whether or not the source of water and toilet facilities used by the mother and child were in good hygienic conditions. Authors further included a feature to represent whether or not households used clean energy for cooking. |

**Table 6. Summary table of predictors, encodings, models, and outcomes.**

| Author | Encoded Features | Encoding Strategy | Model Type(s) | Outcome Definition |
|---|---|---|---|---|
| Nantongo, B. A., et al. [4] | Maternal education, ethnicity, language, ANC visits, place of residence, water source, place of delivery, status of previous vaccination, maternal employment, maternal mobility | Binary | LR, RF, SVM, KNN, XGBoost, Naive Bayes, AdaBoost | Complete vaccination for age |
| | Wealth Index | PCA | | |
| | Maternal age, child age, number of under-5 children | Discretisation | | |
| Hasan, M., et al. [18] | Settlement type, place of residence, maternal education, religious affiliation, media exposure, wealth index, paternal education, maternal employment, sex of child, sex of household head | Binary | Ensemble (GNB, BNB, DT, RF, XGB, LGB) | Receipt of first dose of measles vaccine |
| | Wealth index | PCA | | |
| Demash, A., et al. [19] | maternal education, maternal age, place of residence, sex of child, age of child, sex of household head, place of delivery, maternal employment, media exposure | Binary | Naive Bayes, PART, LR, MLP, J48, Logit Boost, RF, AdaBoost | Complete vaccination for age |
| | Wealth index | PCA | | |
| Biswas, A., et al. [20] | Place of residence, place of delivery, ANC visits, age of the child, child's birth order, sex of child, first pregnancy, maternal education | Binary | Cost-sensitive Ridge | Zero-dose status |
| | Wealth index | PCA | | |
| Mohanraj G., et al. [21] | Maternal education, maternal employment, vaccination card availability, availability of health insurance scheme, paternal employment, ANC visits, sex of child | Binary | Deep Soft Cosine SVM, R-MLP, DT, NB, LR | Complete vaccination for age |
| Abatemam, H., et al. [22] | Maternal education, settlement type, reminders, knowledge of benefits | Binary | Logistic Regression | Complete vaccination for age |
| Abegaz, M., et al. [23] | Maternal age, place of delivery, knowledge of benefits, travel time, Marital status, maternal education, ANC visits, reminders | Binary | Logistic Regression | Complete vaccination for age |
| Muhoza, P., et al. [24] | Maternal age, reminders, settlement type, birth order | Binary | Logistic Regression | Measles 2 and MenA in 2nd year |
| Aheto, J., et al. [25] | Vaccination card, maternal education, vitamin A, access to bank account | Binary | Logistic Regression | Three separate vaccine receipt outcomes |
| Santos, T., et al. [26] | ANC visits, place of delivery, maternal tetanus status | Binary | CART | Zero-dose vaccination |
| | Wealth index | PCA | | |
| Touré, A., et al. [27] | Vaccination card, ANC visits, illness before schedule | Binary | Logistic Regression | Complete vaccination for age |
| Budu, E., et al. [28] | Maternal education, religion, wealth index, settlement type | Binary | Multivariate Stats, LR | Complete vaccination for age |
| Jama, A. [29] | Maternal education, delivery location, travel time | Not Reported | Descriptive Stats | Complete vaccination for age |
| Acharya, K., et al. [30] | Place of residence, sex of child, maternal education | Binary | Logistic Regression | Complete vaccination for age |
| Adamu, A., et al. [31] | Travel time, number of vaccinators, caregiver age | Binary | Logistic Regression, MCMC | Zero-dose vaccination |
| Acharya, P., et al. [32] | Sex of child, place of delivery, preceding birth space of +/- 24 months, maternal education, religious affiliation, maternal employment, media exposure, maternal autonomy, employment status of father, settlement type, travel time, community media exposure rate, community poverty rate, community ANC visit rate, community facility delivery rate, community maternal unemployment rate, community maternal literacy rate, community PNC visit rate. | Binary | Logistic Regression | Complete vaccination for age |
| | Wealth index | PCA | | |

*(Continued)*

**Table 6.** (Continued)

| Author | Encoded Features | Encoding Strategy | Model Type(s) | Outcome Definition |
|---|---|---|---|---|
| Asuman, D., et al. [33] | Maternal age, education, marital status, insurance | Binary | Logistic Regression | Complete vaccination for age |
| | Wealth index | PCA | | |
| Sheikh, N., et al. [34] | Birth season, hygiene, maternal employment, | Binary | Logistic Regression | Complete vaccination for age |
| | Wealth index | PCA | | |

community facility delivery rates as binary representations of high or low. Biswas and colleagues [20] used the one-hot encoding method to transform all data points into binary vector representations. The PCA method was predominantly used to derive a single-point representation for wealth in a number of studies [4,18–20,26]. Also from Table 6, logistic regression was the most commonly employed analytical model, although ensemble machine learning methods such as random forests, gradient boosting machines, and multilayer perceptrons were increasingly evident in studies using secondary survey data. Outcome definitions were primarily binary classifications, such as complete versus incomplete vaccination or zero-dose status.

## Model performance results of machine learning-based studies

In this section we provide insight into the comparative strengths and limitations of various machine learning algorithms for the analysis of childhood vaccination defaulter risk predictors. Table 7 summarizes the reported performance results of models from machine learning-based studies included in this scoping review.

In the study by Mohanraj and colleagues, the R-MLP model reported the highest performance, with a precision of 0.85, recall of 0.78, and accuracy of 0.96, resulting in a high F1 score of 0.81. Though the DT C5.0 achieved the highest recall (0.87), precision was comparably lower (0.72). The LR reported a very low precision of 0.10, resulting in a low F1 score of 0.18. Among the six models tested by Hasan and colleagues, LGB yielded the most optimal performance with a recall of 0.96, precision of 0.80, accuracy of 0.79, and the highest AUC (0.78). RF and XGB also reported relatively high recalls (RF = 96, XGB = 93) and AUC (RF = 0.725, XGB = 0.769). GNB and BNB, though showed comparable high recall (0.95 and 0.96), were affected by much lower precision of 0.76. The study by Nantongo et al., 2024 [4] reported recall very close to 1.00 across most classifiers, with linear models such as LR and SVM, achieving recall ≥ 0.99 and accuracy ≥ 0.94. However, there were significant variations in AUC values. The ensemble classifiers (RF and GBM) reported high AUCs of 0.88, while linear-based models (SVM and LR) reported lower AUCs of 0.45 and 0.55 respectively. The NB classifier also reported very low recall and accuracy of 0.36 but had a relatively high precision of 0.96. The study by Demsash et al., 2023 [19] used a range of algorithms that generally reported strong model performances. The PART algorithm had the highest F1 score of 0.94 and AUC of 0.92. The next models achieving high scores across all metrics were the MP and RF. The J48 model also reported F1 score of 0.78 and AUC of 0.86. However, traditional models such as LR and NB reported relatively low performance in both accuracy and AUC, as observed in the foregoing studies. In the study by Biswas and colleagues [20], very limited metrics were reported on only the CSRC algorithm. The CSRC algorithm reported a recall of 0.80 but a notably low precision of 0.43, indicating a high false-positive rate.

## Discussion

This scoping review highlights the multifaceted nature of predictors associated with childhood vaccination defaulter risk in low-resource settings. From the reviewed studies, there were very marginal differences in how features were encoded, engineered and represented for analysis. Over 90% of the features used were encoded as binary inputs, facilitating ease of analysis in machine learning models. However, while this approach simplifies computation, it may not fully capture the

**Table 7. Model performance results of machine learning-based studies.**

| Study | Algorithm | Recall | Precision | Accuracy | F1-Score | AUC |
|---|---|---|---|---|---|---|
| Mohanraj, G., et al., 2020 [21] | R-MLP | 0.78 | 0.85 | 0.96 | 0.81 | – |
| | DT C5.0 | 0.87 | 0.72 | 0.79 | 0.79 | – |
| | NB | 0.53 | 0.54 | 0.85 | 0.53 | – |
| | LR | 0.68 | 0.10 | 0.88 | 0.18 | – |
| Hasan, M., et al., 2021 [18] | GNB | 0.95 | 0.76 | 0.74 | – | 0.58 |
| | BNB | 0.95 | 0.76 | 0.74 | – | 0.58 |
| | RF | 0.96 | 0.78 | 0.77 | – | 0.73 |
| | DT | 0.79 | 0.81 | 0.71 | – | 0.62 |
| | XGB | 0.93 | 0.81 | 0.78 | – | 0.77 |
| | LGB | 0.96 | 0.80 | 0.79 | – | 0.78 |
| Nantongo, B., et al., 2024 [4] | LR | 1.00 | 0.94 | 0.94 | – | 0.55 |
| | DT | 0.97 | 0.97 | 0.94 | – | 0.74 |
| | RF | 0.99 | 0.96 | 0.96 | – | 0.88 |
| | GBM | 0.99 | 0.97 | 0.96 | – | 0.88 |
| | AdaBoost | 0.99 | 0.97 | 0.96 | – | 0.86 |
| | KNN | 0.99 | 0.94 | 0.94 | – | 0.56 |
| | NB | 0.36 | 0.96 | 0.36 | – | 0.60 |
| | SVM | 1.00 | 0.94 | 0.94 | – | 0.45 |
| | XGBoost | 0.93 | 0.94 | 0.88 | – | 0.45 |
| Demsash, A., et al., 2023 [19] | PART | – | 0.94 | 0.96 | 0.94 | 0.92 |
| | NB | – | 0.74 | 0.66 | 0.68 | 0.72 |
| | RF | – | 0.88 | 0.82 | 0.89 | 0.83 |
| | Logit Boost | – | 0.71 | 0.77 | 0.71 | 0.73 |
| | J48 | – | 0.76 | 0.89 | 0.78 | 0.86 |
| | AdaBoost | – | 0.72 | 0.69 | 0.71 | 0.73 |
| | MP | – | 0.81 | 0.87 | 0.82 | 0.83 |
| | LR | – | 0.72 | 0.66 | 0.71 | 0.66 |
| Biswas, A., et al., 2023 [20] | CSRC | 0.80 | 0.43 | – | 0.87 | – |

complexities of socio-economic and behavioural factors influencing vaccination uptake. For example, predictors such as maternal education and healthcare utilisation, which were consistently significant across studies, were often represented through dichotomous features. Also, the wealth index feature derived from dimensionality reduction techniques, while offering a structured approach to reducing high dimensionality, may inadvertently obscure granular differences within socio-economic features. Aheto, J., et al. [25] tried to circumvent this by distinctively representing some of the unique socio-economic assets, like bank accounts and ownership of livestock. The features present across almost all studies were maternal education/literacy, antenatal care (ANC) visits, and place of delivery. This underscores the pivotal role of maternal literacy and healthcare engagement in determining vaccination defaulter risk. These findings align with those by Acharya and colleagues [30] that maternal education disparities influence vaccination adherence more than economic inequalities alone. Acharya and colleagues [32] also engineered the notion of community-level features, such as community maternal literacy rates, community ANC visit rates, and maternal unemployment rates, which provide a broader socio-structural context to vaccination defaulter risk predictors. From the review, it is apparent that current strategies for identifying predictors of childhood vaccination defaulter risk within the sub-Saharan African regions are predominantly grounded in secondary statistical analyses of retrospective data [22–25,27,32–34,36]. While useful for post hoc

**Table 8. A comparison with previous reviews on childhood vaccination predictors.**

| Aspect | Nour, T., et al. [11] | Galadima, A., et al. [14] | Desalew, A., et al. [13] | Our review |
|---|---|---|---|---|
| Region of Focus | Ethiopia | Sub-Saharan Africa | Ethiopia | Sub-Saharan Africa (focus on Ghana, Nigeria, Ethiopia, etc.) |
| Study Type | Systematic Review | Systematic Review | Meta-analysis | Scoping Review |
| Primary Objective | Identify predictors of vaccination coverage | Identify determinants of vaccine uptake | Assess incomplete vaccination predictors | Explore feature representation and modelling |
| Inclusion of ML Techniques | No | No | No | Yes |
| Discussion of Feature Encoding | No | No | No | Yes |
| Focus on Feature Engineering | No | No | No | Yes |
| Modelling Methods Compared | Regression only | Mostly regression | Regression | Regression & ML (e.g., RF, SVM, MLP) |
| Granularity of Predictor Analysis | General (categorical counts) | General (socio-demographics) | General | Detailed (variable transformation, PCA, composite features) |

evaluations, these approaches offer limited utility for capturing real-time risk dynamics or informing timely interventions. As a result, they fall short in supporting proactive outreach efforts and efficient resource allocation in public health settings. This limitation underscores the growing need for predictive frameworks that can identify defaulter risk prospectively. Nonetheless, we also draw attention to insights worth considering when working with machine learning-based frameworks. Findings from this review suggest that ensemble models such as the Random Forest and different flavours of the Gradient Boosting Machines (GBM, XGB, AdaBoost, LGB) have higher performance tendencies. On the other hand, traditional models such as the Naive Bayes, Support Vector and Logistic Regression models generally showed limited suitability for complex risk prediction tasks, indicated by lower AUCs and poor precision. Also, models that reported perfect recall like LR and SVM in Sheikh, N., et al. he study by Nantongo and colleagues [4] showed low AUCs, which may indicate issues of overfitting or dataset imbalance. Another important finding from the reviews relates to the relatively small data size used by studies that collected new primary data. Indeed, such efforts are tedious to implement and often incur significant costs in terms of both infrastructure and labour to obtain adequate data volumes. The study by Muhoza and colleagues [24] highlighted the possible impacts of data availability on detecting predictors using analytical methods. Data from the Northern region of Ghana (n = 870) identified the largest number of predictors compared to data from the Greater Accra Region (n = 370) and the Volta Region (n = 315).

One major strength of this review in contrast with previous works is the extensive range of studies analysed, which encompass diverse geographical regions and methodological approaches. As a result, it provides a thorough perspective on how predictors are encoded, engineered and represented to analyse vaccination defaulter risk in low-resource settings. We further extend previous works by focusing on how predictors were represented and modelled, particularly in machine learning frameworks. Table 8 contrasts the current review with three cited prior works based on key aspects.

From the results presented, Nour, T., et al. [11]; Desalew, A., et al. [13]; and Galadima, A., et al. [14] conducted comprehensive syntheses of determinants of immunization within the context of sub-Saharan Africa. In all three reviews, the focus was primarily on identifying the determinant of vaccination uptake and coverage. Methods compared mostly involved logistic regression, while the level of analysis was on the counts and frequencies of predictors. However, they did not address the technical aspects of feature encoding or modelling approaches. Our review fills this gap by systematically mapping not just the predictors but also unravelling insights into the details of how the predictors were pre-processed

through engineering and encoding techniques across studies. In doing so, this review offers a methodological advancement that can guide the development of more robust and reproducible machine learning models for vaccination defaulter risk prediction.

Nonetheless, we acknowledge the limitation of potential biases that may arise from the dependence on secondary data sources, which may fail to capture real-time changes in data representation for vaccination predictors. Furthermore, inconsistencies in the definitions of predictors across studies could affect comparability and generalisability. Addressing these limitations in future research will be essential for refining predictive models and enhancing vaccination adherence analysis.

There are relatively high resource commitments from both governments and international bodies for suitable solutions to improve childhood vaccination uptake in low-resource settings. High demands are placed on emerging data-driven techniques that could consider larger information resources for realising deeper insights and possibly making predictions to inform a more reactive intervention process. Knowledge on effective feature engineering and representation can promote the use of machine learning algorithms in the development of real-time dashboards for monitoring vaccination adherence. This can enable timely identification of under-vaccinated populations and prompt interventions before defaulter risks escalate.

## Conclusion

This review synthesises key features for childhood vaccination defaulter risk prediction in low-resource settings, providing a clear indication of feature representation. The results attained from this review provide insights into how best complex features like community-based indicators, migration patterns, and environmental conditions can be represented for machine learning-based analysis and prediction of childhood vaccination defaulter risk. In view of the predominant use of binary encodings, frequency encoding methods for categorical data can be explored. Also, temporal data such as delays in vaccination can be represented as single-point binary features to predict other outcomes of defaulter risk such, as receiving complete dose.

## Supporting information

**S1 PRISMA Checklist. This checklist provides a detailed account of how the reporting standards of the PRISMA 2020 (Preferred Reporting Items for Systematic Reviews and Meta-Analyses) guidelines were adhered to in the conduct of this scoping review.** It includes itemised responses to checklist items covering all key sections of the review. (DOCX)

**S1 Data. This file includes key information for each of the studies reviewed, which include predictors used, outcome definitions, modelling approaches, encoding strategies, and feature engineering notes.** (XLSX)

**S1 Appendix. Supplementary narratives of reviewed studies.** The supplementary narratives contain summaries for each of the studies reviewed: the authors, predictors identified in the articles, the methods used to predict the outcome of interest, and the outcomes of interest predicted. (DOCX)

## Author contributions

**Conceptualization:** Eliezer Ofori Odei-Lartey.

**Data curation:** Eliezer Ofori Odei-Lartey, Stephaney Gyaase, Solomon Nyame.

**Formal analysis:** Eliezer Ofori Odei-Lartey.

**Methodology:** Eliezer Ofori Odei-Lartey, Stephaney Gyaase, Solomon Nyame.

**Supervision:** Dominic Asamoah, Kwaku Poku Asante, James Ben Hayfron-Acquah.

**Writing – original draft:** Eliezer Ofori Odei-Lartey.

**Writing – review & editing:** Stephaney Gyaase, Solomon Nyame, Dominic Asamoah, Kwaku Poku Asante, James Ben Hayfron-Acquah.

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
