## [Decision Letter · Decision Letter 0]

Response to Reviewers
Revised Manuscript with Track Changes
Manuscript
**Journal Requirements:**
**Additional Editor Comments (if provided):**
**Reviewers' Comments:**

**Comments to the Author**

1. Does this manuscript meet PLOS Digital Health’s publication criteria?

Reviewer #1: Yes

Reviewer #2: Yes

2. Has the statistical analysis been performed appropriately and rigorously?

Reviewer #1: N/A

Reviewer #2: N/A

3. Have the authors made all data underlying the findings in their manuscript fully available (please refer to the Data Availability Statement at the start of the manuscript PDF file)?

Reviewer #1: Yes

Reviewer #2: Yes

4. Is the manuscript presented in an intelligible fashion and written in standard English?

Reviewer #1: Yes

Reviewer #2: Yes

Reviewer #1: The manuscript explores a timely and underexamined topic at the intersection of machine learning and public health: the representation, encoding, and engineering of predictors used to assess childhood vaccination defaulter risk in low-resource settings. This is a critical area of research, especially as machine learning applications in public health continue to expand and require greater transparency and rigor in feature selection and preprocessing.

However, in its current form, the manuscript faces several methodological and reporting challenges that limit its clarity, reproducibility, and overall impact.

First, the manuscript lacks transparency in terms of data availability. Although the authors state that data were extracted from publicly available studies, they do not include the full data extraction sheet or a structured summary of the included studies in a machine-readable format (e.g., .csv or .xlsx). For a scoping review, this level of transparency is essential. A detailed supplementary file outlining study characteristics, predictor variables, modeling approaches, and feature encoding strategies should be made available to ensure reproducibility and enable other researchers to build upon the findings.

Second, while the manuscript presents a narrative synthesis of the included studies, it lacks a systematic comparison across them. The addition of a clear summary table mapping predictors to encoding strategies, model types, and outcome definitions would provide much-needed structure and allow for clearer interpretation of patterns and gaps in the literature. Without this, readers are left with descriptive lists that are difficult to synthesize or act upon.

Third, the manuscript states that both statistical and machine learning methods were used for analysis, but the results of these analyses are not presented in the Results section or in the appendices. Even partial findings—such as frequency counts, clustering summaries, or principal components used—would help validate the claims made and enhance the rigor of the review. The absence of this information undermines the strength of the conclusions and leaves the reader uncertain about the analytical contributions of the study.

Reviewer #2: This scoping review is an excellent addition to the literature. It does a great job of providing valuable insights on encoded features to analyze childhood vaccination defaulter risk in low-resource settings. The authors included studies that used statistics or Machine Learning (ML) to predict childhood vaccination defaulter risk, focusing on African countries. According to their findings, there are a variety of common features used between included studies. The authors have found that categorical data were mostly inputted as binary data. However, Primary Component Analysis was used for features that determined socioeconomic status due to their high dimensionality. That said, there are a few minor revisions that should be addressed before progressing to publication.

Minor issues:

1. I suggest reviewing the manuscript for punctuation and grammar errors, particularly in the Results section.

2. In the Methods section, “Search Strategy” paragraph, line 153, I advise specifying the end month (i.e., January 2018) for consistency with the reported end date.

3. There are several instances throughout the manuscript where in-text citations did not mention author names and only included numbers (i.e., line 193 and line 234).

4. In the PRISMA chart (Figure 1), it is not clear that the total number of included articles is 18, as noted in Table 1, line 191, and line 198. Further clarification is needed to distinguish between “New studies included in review n=4” and “Reports of new included studies n=4”. I suggest reviewing the PRISMA chart to ensure consistency with the text.

5. In Table 1, row 4, column 6, there is missing information regarding the outcome of the study by Biswas, A, et al.

6. In Table 2, punctuation should be reviewed as there are a few instances where periods start before the in-text citation.

**Do you want your identity to be public for this peer review?** For information about this choice, including consent withdrawal, please see our Privacy Policy

Reviewer #1: No

Reviewer #2: **Yes: ** Sara Bashar Qasrawi

**Figure resubmission:****Reproducibility:**To enhance the reproducibility of your results, we recommend that authors of applicable studies deposit laboratory protocols in protocols.io, where a protocol can be assigned its own identifier (DOI) such that it can be cited independently in the future. Additionally, PLOS ONE offers an option to publish peer-reviewed clinical study protocols. Read more information on sharing protocols at https://plos.org/protocols?utm_medium=editorial-email&utm_source=authorletters&utm_campaign=protocols

---

## [Editor Report · Decision Letter 1]

Feature Representation in Analysing Childhood Vaccination Defaulter Risk Predictors: A Scoping Review of Studies in Low-Resource Settings

PDIG-D-25-00143R1

Dear  Dr. Eliezer Ofori Odei-Lartey,

We are pleased to inform you that your manuscript 'Feature Representation in Analysing Childhood Vaccination Defaulter Risk Predictors: A Scoping Review of Studies in Low-Resource Settings' has been provisionally accepted for publication in PLOS Digital Health.

Best regards,

Cleva Villanueva, M.D., Ph.D.

Academic Editor

PLOS Digital Health

**Additional Editor Comments (if provided):**

The authors have appropriately addressed all the comments and questions raised by the reviewers and have revised the manuscript accordingly. The manuscript meets the criteria for publication in PLOS Digital Health